# Brief Cognitive Behavioral Therapy for Depression and Anxiety in Patients with Schizophrenia in Psychiatric Home Nursing Service: Pilot Randomized Controlled Trial

**DOI:** 10.3390/bs14080680

**Published:** 2024-08-05

**Authors:** Masayuki Katsushima, Eiji Shimizu

**Affiliations:** 1Department of Rehabilitation, Faculty of Health Care and Medical Sports, Teikyo Heisei University, Ichihara 290-0193, Chiba, Japan; 2Research Center for Child Mental Development, Chiba University, Chiba 260-8670, Chiba, Japan; 3Department of Cognitive Behavioral Physiology, Graduate School of Medicine, Chiba University, Chiba 260-0856, Chiba, Japan; eiji@faculty.chiba-u.jp

**Keywords:** schizophrenia, depression and anxiety, brief cognitive behavioral therapy, psychiatric home nursing, workbook

## Abstract

This pilot randomized controlled trial (RCT) aimed to explore whether depression and anxiety could be reduced by psychiatric home nursing supporters offering brief cognitive behavioral therapy for psychosis (CBTp) at home, using a workbook for patients with schizophrenia. Eighteen patients with schizophrenia treated in a psychiatric home nursing service were randomly assigned to two groups: one group received CBTp in addition to usual care (TAU + CBTp group; n = 9) and the other received only usual care (TAU group; n = 9); two patients were excluded due to hospitalization or withdrawal of consent. Eight weekly CBTp sessions were conducted; anxiety/depression, quality of life, self-esteem, and overall functioning were assessed at baseline, week 9, and week 13. There was no significant difference in the primary and secondary evaluations. The effect size was 0.84 for primary evaluation indicating a large effect. This study showed that nurses and occupational therapists can provide CBTp in psychiatric home nursing for patients with schizophrenia to potentially alleviate anxiety and depression compared to standard psychiatric home nursing service alone. Therefore, larger RCTs with larger sample sizes are recommended.

## 1. Introduction

Clinical evidence has shown that cognitive behavioral therapy (CBT) is effective in treating symptoms of depression and anxiety [1]. CBT has expanded beyond depression and anxiety to include a wide range of psychiatric disorders [2]. Schizophrenia is no exception; since the 1990s, large-scale randomized control trials (RCTs) have been conducted mainly in the U.K.

One in every one hundred people develop schizophrenia [3,4]. Schizophrenia is described as a disorder of the form and content of perception and thought (National Institute of Health and Clinical Excellence (NICE); Psychosis and schizophrenia in adults, 2014), and is associated with significant social or occupational dysfunction [5] in some patients. In contrast, early intervention, appropriate pharmacotherapy, and psychosocial support have been reported to lead to recovery and remission [6].

Conversely, cognitive behavioral therapy of psychosis (CBTp) for schizophrenia has been widely used as a therapeutic intervention since the 1990s, especially in the U.K., and many textbooks have been published on this [7,8,9]. CBTp is used in 58% and 91.3% of the medical institutions in the U.S. and U.K., respectively [10]. Guidelines from the NICE [11] in the U.K. and American Psychiatric Association [12] in the U.S. have been published. A meta-analysis of 33 RCTs (n = 1940) on CBTp reported an effect size of 0.35–0.44 [13].

Although the recommended standard duration of CBTp is approximately 16 sessions or more over 4–6 months, some evidence has been reported for short-term CBTp, consisting of approximately 6–10 sessions over less than 4 months, suggesting that short-term CBTp may have some benefit in treating symptoms of schizophrenia [14,15]. One report found that six brief CBTp sessions conducted by community psychiatric nurses visiting patients in their homes effectively improved depression and illness [16]. Others have reported that six brief CBTp sessions in patients’ homes are effective for patients with low delusional beliefs [17].

Supporters visiting patients in their homes and living environments to provide support and intervention as community life support for people with mental disorders is said to improve their quality of life (QOL) [18]. In contrast, patients with schizophrenia tend to have latent anxiety, depressive symptoms, and low self-esteem due to self-stigma and other factors [19]. These negative and depressive symptoms are also said to reduce their QOL [20]. Turkington et al. reported that CBTp by psychiatric community nurses improves depression and illness awareness [16], but the use of CBTp in psychiatric home nursing is not common in Japan, and there are no RCT studies on CBTp yet. Therefore, we hypothesized that nurses, public health nurses, and occupational therapists involved in psychiatric home nursing could contribute to reducing depression and anxiety in the community life of patients with schizophrenia by providing simple CBTp utilizing a workbook, in addition to their usual home nursing support.

The purpose of this study was to evaluate, by means of a pilot RCT, whether a brief CBTp given by a psychiatric home nursing service to patients with schizophrenia living in the community, compared to usual home nursing, would reduce depression and anxiety.

## 2. Materials and Methods

### 2.1. Study Design

The nurses or occupational therapists recruited two patients from among their own patients through the poster description. We randomly assigned those who agreed to participate in two groups using the envelope method and made parallel group comparisons (Figure 1). We designated the group that received only usual home nursing care as the usual care (TAU) group, and the group that received usual home nursing care in addition to the CBTp intervention, as the TAU + CBTp group. For the intervention group, we conducted CBTp using manual materials during weekly home nursing visits. The nurses or occupational therapists continued to care for both patients in the same charge as before assigned to the two groups, adding CBTp to usual home nursing for patients in the TAU + CBTp group and providing usual home nursing for patients in the TAU group. To compare the groups, we used a scale at each time point: pre-test, a week after the 8-week CBTp intervention period (post-test), and a month after the 13-week follow-up test (FU-test). We used the pre-test and FU test as the baseline and endpoint, respectively, and changes in the baseline and FU test were used as the primary and secondary endpoints.

### 2.2. Sample Size

A total of 24 patients (12 each in the TAU group and 12 in the TAU + CBTp groups) were initially selected as the target sample size as per a previous study [21], which suggested that effect size estimation was possible even with 12 cases in each group in a pilot study.

### 2.3. Inclusion and Exclusion Criteria

Participants were patients diagnosed with schizophrenia as an underlying medical condition, by their primary psychiatrist. They were enrolled at a home healthcare nursing station, a cooperating facility for the study, and after receiving a full explanation of their participation in the study, they fully understood and provided their free and voluntary written consent.

The exclusion criteria were as follows: four consecutive cancellations after the start of the intervention; withdrawal of consent or request to participate in the study; hospitalization during the study period due to worsening psychiatric symptoms; a serious or progressive physical illness; in case of observed suicidal thoughts; an agitated psychomotor state; a history of dependence on or abuse of drugs or alcohol within the past year; inability to give written consent of their own free will with full understanding despite receiving sufficient explanation of their participation in the study. Even if the participants had requested to participate in this study, if the principal investigator determined that continuing the clinical trial would be expected to worsen their psychiatric symptoms, the study would be terminated due to patient safety and ethical considerations.

### 2.4. Intervention

As a psychosocial intervention to improve depression and anxiety, we used “Do-it-yourself cognitive-behavioral therapy” [22] as a workbook to be used in sessions when conducting CBTp. The contents of the sessions were as follows: (1) scoring emotions, (2) self-affirmation, (3) degree of confidence, (4) looking for other thoughts, (5) recording thought changes, (6) noticing patterns of attention, (7) noticing a counterexample, and (8) activating actions. There was a total of eight sessions, each lasting approximately 40 min, conducted in the patients’ homes through visits at all times by in-charge nurses and occupational therapists who were skilled psychiatrists with more than 10 years of experience in clinical psychiatry. They additionally received approximately 10 h of pre-session training from the study administrator (the author) and supervision after each session.

The TAU group was provided with eight home nursing visits as usual, with the same frequency over an eight-week period and the same duration as the TAU + CBTp group; participants in the TAU and TAU + CBTp groups received home nursing visits from the same primary support person prior to study entry.

### 2.5. Outcome Measures

Information on age, gender, number of hospitalizations, age at onset, presence of a cohabitant, and amount of antipsychotic medication being taken were collected from medical records at baseline. The patients were evaluated at pre-test (baseline), post-test, and FU tests (endpoints) for the primary and secondary outcomes.

#### 2.5.1. Primary Outcome

Kessler Psychological Distress Scale (K6; Japanese version) was used as the primary endpoint; the K6 self-administered questionnaire screens for anxiety and depressive symptoms [23]. The Japanese version is a self-administered rating scale created by Furukawa et al. that has been examined for reliability and validity [24]. The questionnaire is very simple, consisting of only six questions, and is not burdensome for the examinee. The K6 has been found to be useful in serious illnesses such as schizophrenia because of its simplicity [25]. To our knowledge, there is no literature on the use of the K6 as a primary endpoint in schizophrenia intervention studies; however, it has been developed for community mental health research that appropriately reflects distress and psychological anxiety and is useful for monitoring mental health status [26] and has shown versatility in monitoring depression and anxiety among community residents [27]. In addition, patients with schizophrenia have some cognitive impairment [28], and the K6 is considered suitable for participants to work with as a method of assessing anxiety and depression in patients with schizophrenia supported by psychiatric home care because of its six questions and the plain and easy to understand wordings of the questions.

#### 2.5.2. Secondary Outcomes

We used the following secondary outcomes.

Beck Depression Inventory-Second Edition (BDI-II). Depression severity was assessed using the Japanese version [29] of the BDI-II [30], a self-administered questionnaire that evaluates depression severity over the past two weeks. It comprises 21 questions in total, with a score of 13 or less indicating “very mild depression”, 14 or more “mild depression”, 20 or more “moderate depression”, and 29 or more “severe depression”. The maximum possible score is 63.

Schizophrenia Quality of Life Scale-Japanese version (JSQLS). The JSQLS was created as a Japanese version [31] and its reliability and validity have been examined. The original version, the Schizophrenia Quality of Life Scale (SQLS) [32] is a QOL assessment scale for patients with schizophrenia. In this study, the JSQLS was used to evaluate participants’ QOL. It consists of a total of 30 questions, with sub-items in the three domains of “motivation and vitality”, “symptoms and side effects”, and “psychosocial relationships”, each of which can be compared on a 100-point scale. The total score is 120 points on a 5-point scale ranging from “never” to “always”, with higher scores denoting poorer QOL.

Rosenberg Self-Esteem Scale-Japanese version (RSES-J). Self-esteem was measured using the RSES-J [33], which was developed as a Japanese version of the original RSES [34], a 4-item self-administered questionnaire with each question scored on a scale of 1 to 4. The higher the total score, the higher the self-esteem. The total score ranges from 10 to 40 points.

Global Assessment of Functioning (GAF). The GAF, which is used to assess both the severity of psychiatric symptoms and the level of functioning, is a rating of overall functioning and is quantified on a 100-point scale. Its reliability and validity have been examined [35]. A home health nurse and occupational therapist observed the participants and scored their assessments.

### 2.6. Data Analysis

Due to the small sample size, we tested whether the TAU + CBTp and TAU groups for each outcome were normally distributed using the Shapiro–Wilk normality test. The results showed that the TAU + CBTp group was found to be normal for all outcomes. However, normality was rejected for K6, BDI-II, and GAF in the TAU group. Therefore, a nonparametric statistical method, Mann–Whitney’s U test, was used for the comparison between the two groups. For the TAU + CBTp and TAU groups, patient background attributes were compared using Fisher’s exact test. For continuous and categorical variables, respectively, the baseline values (pre-test) for each rating scale were compared between the two groups using Mann–Whitney U test.

For outcomes, Mann–Whitney U test was performed to determine the amount of change from baseline. Effect sizes (hedge’s *g*) for the pre-test–Fu-test interval for the TAU + CBTp group were also estimated for reference. Absolute values of 0.2 to less than 0.5, 0.5 to less than 0.8, and 0.8 or greater indicate small, moderate, and large effect sizes, respectively [36].

As a sensitivity analysis, a repeated measures mixed-effects model was used to examine the differences between the TAU + CBTp and TAU groups in terms of their rating scale scores at each time point (pre-test, post-test, and FU-test), the interaction between the groups and time points (pre-test, post-test, and FU-test), and group and time points as fixed effects. The *p*-values were set at 0.05, two-tailed on both sides. Statistical analyses were performed using IBM SPSS Statistics for Windows, version 29.

### 2.7. Research Ethics

This study was approved in 2014 by the Ethics Review Committees of Chiba University Graduate School of Medicine and Teikyo Heisei University (Approval Nos. 926 and 26-080, respectively). Written and oral explanations were provided to the collaborating institutions and study participants, and their written consent was obtained, stating that they could withdraw their participation from the study at any point in time and that there would be no disadvantages resulting from withdrawal.

## 3. Results

### 3.1. Participants’ Demographic Information

This study was conducted between November 2014 and November 2016 with 18 participants. As one participant from the intervention group withdrew due to withdrawal of her consent, and one participant from the control group was hospitalized due to deterioration of his condition, there were finally eight participants each in both groups, that is a total of sixteen participants (mean age 47.94 years, standard deviation ± 13.17), were included in the analysis as a per protocol set. There were no significant differences between the two groups in terms of gender, age, age at schizophrenia onset, number of previous hospitalizations, duration of most recent community living (months), or amount of antipsychotic medication (chlorpromazine equivalent) taken at study entry (Table 1). In addition, a comparison of the main and secondary items at baseline (pretest) using a *t*-test showed no significant differences between the two groups.

### 3.2. Primary Outcome

For the primary endpoint, change from baseline in K6, the intervention group was −2.5 points and the control group was 1.0 points, for a between-group difference of 4.5 points (Table 2). Although the mean change was better in the intervention group, the difference was not significant (*p* = 0.07) according to Mann–Whitney’s U test (Table 3). The effect size of K6 at baseline on the endpoint was 0.84 (CI −2.23–1.81), a large effect size.

### 3.3. Secondary Outcomes

As for changes in other endpoints, BDI-II showed a difference between groups of 12.5, but the difference was not significant (*p* = 0.10); JSQLS showed a difference between groups of 14.0, but the difference was not significant (*p* = 0.05); and 2.0 in RESE-J, the TAU group showed more improvement than the TAU + CBTp group and the difference was not significant (*p* = 1.00). The change in GAF was 15.0, which was not significant (*p* = 0.06). The effect sizes for each endpoint at baseline were 0.30 (CI = −0.70–1.27) for BDI-II, 0.26 (CI = −0.74–1.23) for JSQLS, and 0.16 (−1.13–0.83) for RSES-J, and 0.25 (−1.22–0.75).

## 4. Discussion

In this study, we speculated that the CBTp intervention may have contributed in some way to the reduction of anxiety and depression since the K6 score decreased compared to standard psychiatric home nursing alone. Specific findings are discussed below.

### 4.1. Depression and Anxiety

A relatively large effect size was observed for change in K6, although there was no significant difference in the amount of change from baseline to endpoint, one month after completion of the 8-session CBT intervention between the intervention and the control group. The higher effect size in this study compared to the overall effect size previously reported in a meta-analysis [13] suggests that CBT may also reduce depression and anxiety in patients with schizophrenia. However, the findings should be interpreted with caution since the sample size was limited because it was a pilot study.

### 4.2. Quality of Life

Previous studies have reported that improving coping skills for depression and anxiety improves quality of life because it leads to self-help [37] and CBTp interventions may contribute to improved coping skills [38]. However, in the current study, there were no significant improvements in JSQLS.

### 4.3. Possibility of Relapse Prevention

Regarding the change from baseline in the primary and secondary endpoints at the endpoint (FU-test time point), there was a trend toward improvement in the TAU + CBTp group, but worsening in the TAU group for the K6, BDI-II, JSQLS, and GAF endpoints; a trend toward worsening in the K6, BDI-II, and JSQLS in patients in the TAU group may also be related to the likelihood of symptom relapse [39].

However, the intervention group did not experience exacerbations and showed a trend toward improvement, suggesting that the CBTp intervention may have contributed to the prevention of exacerbations. Relapse and rehospitalization are characteristic features of schizophrenia, and prevention of relapse is a major objective of community life support. There have been prior reports on stepped-care models for depression and anxiety disorders [11]. A similar model has been identified to be necessary for schizophrenia [40,41].

### 4.4. Implementation Costs

The CBTp may face practitioner shortage [42]; hence, the use of workbook materials may allow CBTp to be practiced with less cost and effort, thereby improving patient access to CBTp. Such CBTp has been reported to have the advantage of being inexpensive, flexible, and easy to use [43]. They may be appropriate for those with access difficulties and can be relatively easily incorporated into the tiered care model of CBTp. Brief cognitive behavioral therapy provided by healthcare professionals other than psychologists with some training was considered to contribute to stepped care.

### 4.5. Limitations and Future Challenges

This study has some limitations. First, the envelope method was used to randomly assign patients to two groups for simplicity of conduction in a home nursing setting; however, the endpoints caused differences between the groups at baseline. In the subsequent RCTs, stratified allocation with respect to baseline severity in both groups may be performed using a computer. Second, objective assessments such as the GAF should be blinded to the assessors to eliminate bias. Third, although the present study used the K6 self-administered scale for depression and anxiety, the use of structured interview rating scales should be considered. Fourth, the sample size was small. Although this was a pilot study, the target sample size of 24 participants in both groups, which was the goal of this study, was not achieved because of the limited duration of the study. A larger study should be conducted in the future.

## 5. Conclusions

In this pilot RCT, the addition of CBTp by nurses, occupational therapists, and other psychiatric home nursing services reduced anxiety and depression scores in patients with schizophrenia compared to usual psychiatric home nursing services alone, but no significant differences were identified.

## Figures and Tables

**Figure 1 behavsci-14-00680-f001:**
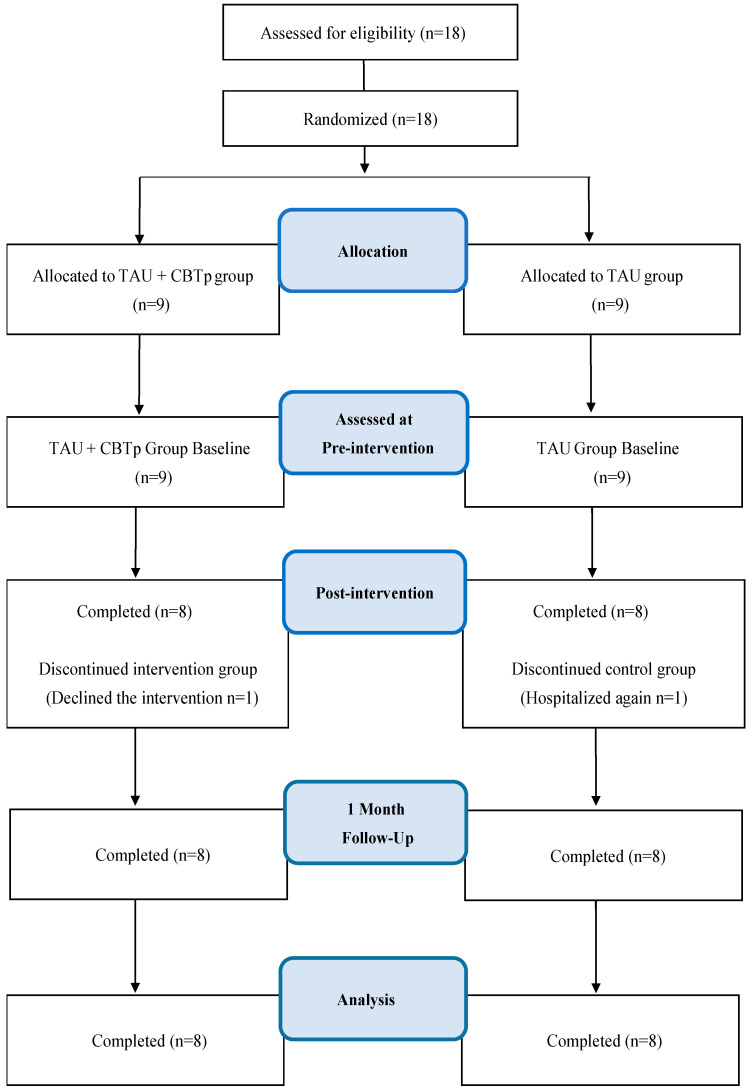
Flow diagram of the recruitment and retention participants in the evaluation.

**Table 1 behavsci-14-00680-t001:** Participant characteristics.

Variables	TAU + CBTp Mean (SD) (n = 8)	TAU Mean (SD) (n = 8)	*p*
Age, years	48.8 (17.3)	49.1 (8.2)	0.70
Men, number (%)	6 (75)	3 (37.5)	0.32
Experience of psychiatric hospitalization, number (%)	7 (87.5)	7 (87.5)	1.00
Hospitalization, number of times	4.1 (3.8)	3.3 (1.9)	0.96
Non-hospitalized period, months	191 (335)	71 (132.5)	0.72
Age of onset, years	24.5 (14.3)	29.1 (11.4)	0.16
Living with parents, number (%)	3 (37.5)	3 (37.5)	1.00
Public participation, number (%)	7 (87.5)	7 (87.5)	1.00
Antipsychotics	901.9 (461.5)	738.2 (327.9)	0.28
(Chlorpromazine equivalent, mg/day)			
	**TAU + CBTp** **Median (IQR)**	**TAU** **Median (IQR)**	** *p* **
K6	5.5 (4.3–7.0)	6.5 (2.8–8.0)	0.44
BDI-II	12.0 (8.3–14.3)	17.0 (4.5–24.0)	0.38
JSQLS	49.0 (33.8–55.3)	50.0 (16.0–61.5)	0.72
RSES-J	26.5 (22.8–29.3)	24.0 (21.8–26.3)	0.44
GAF	53.0 (41.8–60.3)	50.5 (41.0–64.3)	0.96

CBTp: cognitive behavioral therapy of psychosis; TAU: treatment as usual; SD: standard deviation; K6: Kessler’s psychological distress scale; BDI-II: Beck Depression Inventory-Second Edition; JSQLS: Schizophrenia Quality of Life Scale-Japanese version; RSES-J: Rosenberg Self-Esteem Scale-Japanese version; GAF: Global Assessment of Functioning.

**Table 2 behavsci-14-00680-t002:** Raw data in primary outcomes and secondary outcomes.

	TAU + CBTp (n = 8)			TAU (n = 8)		
	Pre	Post	FU	Pre	Post	FU
	Median (IQR)	Median (IQR)	Median (IQR)	Median (IQR)	Median (IQR)	Median (IQR)
K6	5.5 (4.3–7.0)	4.5 (1.5–6.3)	3.0 (1.5–4.3)	6.5 (2.8–8.0)	6.0 (2.8–8.3)	7.5 (1.8–12.0)
BDI-II	12.0 (8.3–14.3)	9.0 (2.3–14.3)	7.5 (4.5–13.3)	17.0 (4.5–24.0)	13.0 (8.0–22.3)	19.5 (6.5–32.8)
JSQLS	49.0 (33.8–55.3)	46.5 (31.5–47.8)	41.5 (29.0–48.0)	50.0 (16.0–61.5)	52.5 (25.3–58.5)	55.5 (33.5–62.5)
RSES-J	26.5 (22.8–29.3)	27.5 (24.3–30.0)	25.5 (24.8–29.8)	24.0 (21.8–26.3)	23.0 (22.0–26.3)	23.5 (22.5–28.5)
GAF	53.0 (41.8–60.3)	60.0 (46.0–65.8)	60.0 (49.0–65.3)	50.5 (41.0–64.3)	49.5 (30.8–61.0)	45.0 (33.8–55.3)

IQR: interquartile range; K6: Kessler’s psychological distress scale; BDI-II: Beck Depression Inventory-Second Edition; JSQLS: Schizophrenia Quality of Life Scale-Japanese version; RSES-J: Rosenberg Self-Esteem Scale-Japanese version; GAF: Global Assessment of Functioning.

**Table 3 behavsci-14-00680-t003:** Amount of change from baseline (pre-test) to endpoint (FU-test) and effect size.

	TAU + CBTp Median (IQR)	TAU Median (IQR)	U	z	*p*	TAU + CBTp Hedge’s *g* (95% CI)
K6	−2.5 (−4.0–0.0)	−0.5 (−1.3–4.8)	49.000	1.803	0.07	0.84 (−0.23–1.81)
BDI-II	−1.0 (−5.3–0.5)	1.5 (−1.0–8.8)	47.500	1.641	0.10	0.30 (−0.70–1.27)
JSQLS	−5.0 (−6.0–1.3)	15 (2.3–18.8)	50.500	1.952	0.05	0.26 (−0.74–1.23)
RSES-J	0.0 (−0.3–1.3)	0.5 (−1.3–3.8)	32.000	0.000	1.00	0.16 (−1.13–0.83)
GAF	5.0 (0.3–9.5)	−0.5 (−4.8–0.0)	14.500	0.156	0.06	0.25 (−1.22–0.75)

Mann–Whitney U test. IQR: interquartile range; Hedge’s g: effect size; CI: confidence interval: K6: Kessler’s psychological distress scale; BDI-II: Beck Depression Inventory-Second Edition; JSQLS: Schizophrenia Quality of Life Scale-Japanese version; RSES-J: Rosenberg Self-Esteem Scale-Japanese version; GAF: Global Assessment of Functioning.

## Data Availability

The data are available upon reasonable request.

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
