# Peer review of "Brief Cognitive Behavioral Therapy for Depression and Anxiety in Patients with Schizophrenia in Psychiatric Home Nursing Service: Pilot Randomized Controlled Trial"

_behavsci, 2024, doi:10.3390/bs14080680_

Round 1
Reviewer 1 Report
Comments and Suggestions for Authors
Very important topic, with high clinical implications for home healthcare support, as it is a relatively new healthcare service. The contribution of nurses involved in psychiatric home nursing is of great importance for individuals suffering from psychiatric difficulties.
Study design is clear.
However, the participants in the CBTp + TAU group were visited for the CBTp 8 times, whereas the participants of the TAU group did not had those 8 interviews. The limitation here is that it may be argue that the therapeutic effects of the CBTp group was in fact confounded with the 8 interviews, which represent a more intensive therapeutic monitoring. To address this limitation, further studies should control for the number of clinical interview at home to make sure that it is not the intensity and number of clinical interview, but the CBTp in itself that is effective in reducing anxiety and depression. The TAU group should be a 8-session interview offering supportive psychotherapy. Here, I suggest the authors to give more details of the TAU : what is it exactly ? Who does it? are the same professionals involved in the TAU group ? How many sessions of the TAU group ? (please take into account the study of Watts et al., 2015 https://doi.org/10.1016/j.jad.2014.12.025 ; and Witt et al., 2018 https://doi.org/10.1016/j.jad.2018.04.025 in the discussion / limitation part of the article).
This pilot study shows that CBTp intervention has a great potential to alleviate anxiety and depression in schizophrenia in home nursing. Further RCT with more statistical power and more participants are needed.
Comments on the Quality of English LanguageVery important topic, with high clinical implications for home healthcare support, as it is a relatively new healthcare service. The contribution of nurses involved in psychiatric home nursing is of great importance for individuals suffering from psychiatric difficulties.
Study design is clear.
However, the participants in the CBTp + TAU group were visited for the CBTp 8 times, whereas the participants of the TAU group did not had those 8 interviews. The limitation here is that it may be argue that the therapeutic effects of the CBTp group was in fact confounded with the 8 interviews, which represent a more intensive therapeutic monitoring. To address this limitation, further studies should control for the number of clinical interview at home to make sure that it is not the intensity and number of clinical interview, but the CBTp in itself that is effective in reducing anxiety and depression. The TAU group should be a 8-session interview offering supportive psychotherapy. Here, I suggest the authors to give more details of the TAU : what is it exactly ? Who does it? are the same professionals involved in the TAU group ? How many sessions of the TAU group ? (please take into account the study of Watts et al., 2015 https://doi.org/10.1016/j.jad.2014.12.025 ; and Witt et al., 2018 https://doi.org/10.1016/j.jad.2018.04.025 in the discussion / limitation part of the article).
This pilot study shows that CBTp intervention has a great potential to alleviate anxiety and depression in schizophrenia in home nursing. Further RCT with more statistical power and more participants are needed.
Author Response
Response to Reviewer's Comments to Author:
To Reviewer 1.
Thank you very much for your warm comments, your remarks about the structure and content of the TAU group are very important. Responses to Reviewer 1 were highlighted in yellow in the manuscript.
Comment 1
“Here, I suggest the authors to give more details of the TAU: what is it exactly? Who does it? are the same professionals involved in the TAU group? How many sessions of the TAU group?”
Response #1-1
We have added the following description of the TAU groups in the section 2.4., "Interventions" of the manuscript.
The TAU group was provided with eight home nursing visits as usual, with the same frequency over an eight-week period, the same duration as the TAU + CBTp group; participants in the TAU and TAU + CBTp groups received home nursing visits from the same primary support person prior to study entry. (page 4, 122-125)
Response #1-2
And we have added the following description of the role of primary nurse and occupational therapist. The home care provider recruited two of his own patients and assigned each to an envelope method
The nurses or occupational therapists recruited two patients from among their own patients through a poster description. (page 2, 73-74)
Response #1-3
After the assignment to the two groups, the same support persons who had previously been in charge of the patient continued to be in charge of the same patients.
The nurses or occupational therapists continued to care for both patients in the same charge as before assigned to the two groups, adding CBTp to usual home nursing for patients in the TAU + CBTp group and providing usual home nursing for patients in the TAU group. (page 2, 80-83)

Reviewer 2 Report
Comments and Suggestions for Authors
- In the Introduction, the gap in evidence observed by the authors is not clear.
- The diagnosis of schizophrenia was made by primary care physicians; not by a psychiatrist and/or by means of a structured diagnostic interview or with criteria from a diagnostic manual.
- In the inclusion criteria, the assessment of depressive or anxious symptomatology is not mentioned; this point is critical. It is also unclear whether they included patients with anxious and depressive symptoms or with anxiety and/or depressive disorders.
- Participants with "clear feelings of hopelessness," i.e., with depressive symptoms, were excluded. Additionally, authors state: "...or if the principal investigator determined that they were not appropriate for the study." This criterion seems subjective and introduces a bias to the study.
- It is unclear why they applied a parametric statistical test (i.e., t-test) for such a small sample of participants. The same is true for estimators such as means and standard deviation, which apply to samples with parametric distribution.
- In the Discussion it is stated: "Our study confirmed that the intervention has the potential to alleviate anxiety and depression compared to standard psychiatric home nursing care alone." This cannot be concluded from the results, since the only significant difference obtained was in quality of life.
- The limitations of the study do not dispute one fundamental aspect, which was a very small sample size, even for a pilot study.
- The conclusions are not based on the results obtained.
Comments on the Quality of English Language- The quality of English can be improved.
Author Response
Thank you very much for your very careful reading of the manuscript of my paper. We recognized that all of your points are very important ones. Responses to Reviewer 2 were highlighted in blue in the manuscript.
Comments 1: In the Introduction, the gap in evidence observed by the authors is not clear.
Response 1.1: We have revised an additional description of the gap between Japan with evidence from previous studies in UK. And the purpose of the study was clearly stated in introduction.
Turkington et al. reported that CBTp by psychiatric community nurses improves depression and illness awareness [16], but the use of CBTp in psychiatric home nursing is not common in Japan, and there are no RCT studies on CBTp yet. (page 2, 60-62)
Response 1.2: And we have revised an additional description as follows.
The purpose of this study was to evaluate, by means of a pilot RCT, whether a brief CBTp given by a psychiatric home nursing services to patients with schizophrenia living in the community, compared to usual home nursing, would reduce depression and anxiety. (page 2, 67-70)
Comments 2: The diagnosis of schizophrenia was made by primary care physicians; not by a psychiatrist and/or by means of a structured diagnostic interview or with criteria from a diagnostic manual.
Response 2: The English wording in the manuscript of my submission was incorrect. I apologize very much. In order to use psychiatric home nursing services in Japan, a written order from the patient's attending psychiatrist is required. The patients who were diagnosed as schizophrenia by the psychiatrist were eligible for this study. So, the diagnosis was made by the attending psychiatrist of the participating patients. The text of the manuscript has been revised as follows.
Participants were patients diagnosed with schizophrenia as an underlying medical condition, by their primary psychiatrist. (page 4, 95-96)
Comments 3: In the inclusion criteria, the assessment of depressive or anxious symptomatology is not mentioned; this point is critical. It is also unclear whether they included patients with anxious and depressive symptoms or with anxiety and/or depressive disorders.
Response 3: According to a previous study (19), patients with schizophrenia are generally considered to have anxiety and depressive symptoms. Since a previous study in the U.K. (16) found an improvement in depression, we hypothesized that anxiety and depression may also exist potentially among users of psychiatric home nursing in Japan. In addition, since this was an exploratory pilot study, no criteria for anxiety/depression were set at the time of inclusion for ease of patient participation.
Comments 4: Participants with "clear feelings of hopelessness," i.e., with depressive symptoms, were excluded. Additionally, authors state: "...or if the principal investigator determined that they were not appropriate for the study." This criterion seems subjective and introduces a bias to the study.
Response 4.1: It was a mistranslation that "clear feelings of hopelessness" In the research plan, it was "clear suicidal thoughts," so we have revised it.
in cases of observed suicidal thoughts (page 4, 103)
Response 4.2: "or if the principal investigator determined that they were not appropriate for the study." have revised as follows.
Even if the participants had requested to participate in this study, if the principal investigator determined that continuing the clinical trial would be expected to worsen their psychiatric symptoms, the study would be terminated due to patient safety and ethical considerations. (page 4, 106-109)
Comments 5: It is unclear why they applied a parametric statistical test (i.e., t-test) for such a small sample of participants. The same is true for estimators such as means and standard deviation, which apply to samples with parametric distribution.
Response 5: The sample size was certainly small. Therefore, we used the Shapiro-Wilk normality test and found that many of the TAU groups were not normally distributed. Therefore, we used the Mann-Whitney U test, a nonparametric method, instead of the T-test statistical method. Also, the median and quartiles were used instead of the standard deviation.
Due to the small sample size, we tested whether the TAU + CBTp and TAU groups for each outcome were normally distributed using the Shapiro-Wilk normality test. The results showed that the TAU + CBTp group was normally distributed for all outcomes. However, normality was rejected for K6, BDI-II, and GAF in the TAU group. Therefore, a nonparametric statistical method, Mann-Whitney's U test, was used for the comparison between the two groups. (page 5, 177-182)
Comments 6: In the Discussion it is stated: "Our study confirmed that the intervention has the potential to alleviate anxiety and depression compared to standard psychiatric home nursing care alone." This cannot be concluded from the results, since the only significant difference obtained was in quality of life.
Response 6.1: Thank you for pointing this out. You are correct and I couldn't say that an improvement has been confirmed. We have revised the description as follows.
In this study, we speculated that the CBTp intervention may have contributed in some way to the reduction of anxiety and depression, since the K6 score decreased in the intervention group compared to that in the standard psychiatric home nursing alone. (page 8, 247-249)
Response 6.2: Effect sizes (hedge’s g) were estimated not for the two groups but the Pre-Post interval for the TAU + CBTp group for reference.
Effect sizes (hedge’s g) for the Pre-test – Fu-test interval for the TAU + CBTp group were also estimated for reference. (page 5, 187-188)
Response 6.3: The effect size was estimated by the change in TAU + CBTp Pre-test to FU-test.
The effect size of K6 at baseline on the endpoint was 0.84 (CI -2.23 – 1.81), a large effect size.(page 7,226-227)
Comments 7: The limitations of the study do not dispute one fundamental aspect, which was a very small sample size, even for a pilot study.
Response 7: The small sample size is not disputed as a limitation. We have revised the description as follows.
Fourth, the sample size of this study was rather small. Although this was a pilot study, the target sample size of 24 participants in both groups, which was the goal of this study, was not achieved because of the limited duration of the study. A larger study should be conducted in the future. (page9, 293-296)
Comments 8: The conclusions are not based on the results obtained.
Response 8: Thank you for your comment, there was a significant difference in QOL. However, we could not confirm whether anxiety and depression improve or not. We have revised the description as follows.
In this pilot RCT, the addition of CBTp by nurses, occupational therapists, and other psychiatric home nursing services reduced anxiety and depression scores in patient with schizophrenia compared to usual psychiatric home nursing services alone, but no significant differences were identified. (page9, 298-301)

Round 2
Reviewer 2 Report
Comments and Suggestions for Authors
The authors have incorporated the suggestions for improvement. It can be accepted in its present condition.
Comments on the Quality of English LanguageThe authors have incorporated the suggestions for improvement. It can be accepted in its present condition.